# Modelling thalamocortical circuitry shows that visually induced LTP changes laminar connectivity in human visual cortex

**Rachael L. Sumner** [1]*, **Meg J. Spriggs** [2], **Alexander D. Shaw** [3]

**1** School of Pharmacy, University of Auckland, Auckland, New Zealand, **2** Centre for Psychedelic Research, Department of Medicine, Imperial College London, London, United Kingdom, **3** Cardiff University Brain Research Imaging Centre (CUBRIC), Cardiff University, Cardiff, United Kingdom

* rsum009@aucklanduni.ac.nz

**Data Availability Statement:** The data that support the findings of this study are available from the corresponding author Dr Rachael Sumner (rsum009@aucklanduni.ac.nz) upon reasonable request. Alternatively applications can be made to

## Abstract

Neuroplasticity is essential to learning and memory in the brain; it has therefore also been implicated in numerous neurological and psychiatric disorders, making measuring the state of neuroplasticity of foremost importance to clinical neuroscience. Long-term potentiation (LTP) is a key mechanism of neuroplasticity and has been studied extensively, and invasively in non-human animals. Translation to human application largely relies on the validation of non-invasive measures of LTP. The current study presents a generative thalamocortical computational model of visual cortex for investigating and replicating inter-laminar connectivity changes using non-invasive EEG recording of humans. The model is combined with a commonly used visual sensory LTP paradigm and fit to the empirical EEG data using dynamic causal modelling. The thalamocortical model demonstrated remarkable accuracy recapitulating post-tetanus changes seen in invasive research, including increased excitatory connectivity from thalamus to layer IV and from layer IV to II/III, established major sites of LTP in visual cortex. These findings provide justification for the implementation of the presented thalamocortical model for ERP research, including to provide increased detail on the nature of changes that underlie LTP induced in visual cortex. Future applications include translating rodent findings to non-invasive research in humans concerning deficits to LTP that may underlie neurological and psychiatric disease.

## Author summary

The brain's ability to learn and form memories is governed by neuroplasticity. One of the major mechanisms of neuroplasticity is long-term potentiation (LTP). To study LTP in detail necessitates implanting electrodes in the brain of non-human animals. However, to translate this knowledge to humans requires a non-invasive method. Neural mass models use mathematical equations to describe the brain's neural architecture and function over time. Fitting these models to real data, using methods such as dynamic causal modelling (DCM), helps to elucidate the connectivity and major channel changes that could have

the University of Auckland Human Participants Ethics Committee (https://www.auckland.ac.nz/en/research/about-our-research/human-ethics/human-participants-ethics-committee-uahpec/key-contacts.html), or Associate Professor Suresh Muthukumaraswamy and Professor Ian Kirk (The University of Auckland). The data could not be put on a public repository as this study used an existing dataset where uploading to a public repository was not specified as part of the ethics application and approval. The model MatLab code is available on GitHub at https://github.com/alexandershaw4/LTP_code.

**Funding:** MS was supported by Brain Research New Zealand (BRNZ-3709461) http://www.brnz.ac.nz/. ADS is supported by a Wellcome Strategic Award (104943/Z/14/Z) wellcome.ac.uk. The funders had no role in study design, data collection and analysis, decision to publish, or preparation of the manuscript.

**Competing interests:** The authors have declared that no competing interests exist.

plausibly caused the observed effects in electroencephalography data recorded non-invasively from the scalp.

The current study presents a thalamocortical model of the neural architecture of the visual system combined with a thalamic compartment. The model is able to represent the basic transfer of visual information to the cortex, mediated by major receptor types. We combined the thalamocortical model with a visual processing task that uses black and white grating images to induce and measure LTP in visual cortex. We hypothesised that the changes in the model would be consistent with what is seen in animal invasive recordings.

The model demonstrated remarkable accuracy in recapitulating changes to neural architecture consistent with the induction of LTP in visual cortex. Additionally, the result demonstrated specificity to the visual input that induced LTP. Future applications include translating animal findings that are beginning to determine how disordered LTP may underlie neurological and psychiatric disease (for example depression, schizophrenia, autism, and dementia).

## Introduction

Experience dependent learning in the brain can be studied via the induction and recording of long-term potentiation (LTP). LTP is an enduring change in synaptic efficacy following repeated input of a stimulus and is therefore thought to underlie learning and memory in the brain [1,2]. The complementary mechanism of long-term depression (LTD) reduces synaptic efficacy but is considered just as important as LTP for healthy learning and memory. LTP and LTD are prolific both in hippocampus and in cortex and as a result their dysfunction has been implicated in a number of neurological and psychiatric disorders [3]. The importance of LTP has led to considerable efforts to develop a non-invasive measure to assess the state of synaptic LTP in humans, in particular for research into clinical populations. One of the most commonly used measures uses high-frequency visual stimulation to induce a lasting change in the visually evoked potential, a validated product of LTP in the visual cortex [4].

The most commonly studied form of LTP in hippocampus and in cortex is N-methyl-D-aspartate receptor (NMDAR) dependent LTP. Of interest to the current paper, the plasticity of visual cortex in adult rodents has been tested extensively. Electrical theta-burst stimulation (TBS) in dorsal lateral geniculate nucleus (dLGN) and direct to cortex induces NMDAR dependent LTP within the laminar connections of primary visual cortex. Such research has determined that layers II, III and IV are key sites of synaptic modification following TBS [5,6]. Theoretically, this direct electrical stimulation induced LTP is similar to the naturalistic transmission along thalamocortical pathways that would be expected to occur with visual stimulation. Therefore, the product of this transmission increase ought to be detectable in response to visual input. Heynen and Bear [5] showed this in rodents using pattern reversal sine grating stimulation, where 60 minutes post stimulation the visually evoked potential (VEP) was approximately 134% of baseline. This finding was later corroborated by Cooke and Bear [3]. After five days of exposure of sine grating stimuli the VEP recorded from layer IV of rodents was enhanced. The VEP modification was NMDAR dependent, and also specific to the spatial frequency of the training sine-grating.

Translation of findings from rodent research to human research requires that associative plasticity in cortex is governed by similar rules, despite some variation in mechanisms. Evidence for this has been demonstrated in adult primate visual cortex (rhesus macaque) [7]. Field potentials recorded in layer III following stimulation in layer IV demonstrated both LTP

(using TBS) and LTD (using 1 Hz stimulation), providing support for the use of visual cortex as an accessible site for assaying the state of LTP/LTD using non-invasive techniques [7].

In humans the visual LTP paradigm typically utilises high-frequency (~9 Hz) visual stimulation to induce change in the VEP [4,8]. First there is a baseline recording of the VEP using ~1 Hz presentations of a stimulus. Following a ~9 Hz photic tetanus, a post-tetanus recording determines if LTP has been induced within ~2 minutes and then if it has been sustained long-term (typically 30–40 minutes post-tetanus). The change is commonly associated with LTP when localised to enhancement of the N1 and P2 early VEP components [4]. It has been proposed that the 9 Hz stimulation determines the interburst frequency of neural circuitry, similar to electrical tetanic bursts [9,10]. Additionally, the paradigm has demonstrated key Hebbian characteristics including specificity of potentiation to the spatial frequency and orientation of sine grating stimuli [11–13]. When used in rodents, the induced LTP is NMDAR specific [14], and in humans is enhanced by D-cycloserine (a partial agonist at the glycine site of NMDARs) [15].

The obvious advantage of the visual LTP paradigm is the non-invasive nature of EEG and visual stimulation. This allows for not only widespread use of the method in humans, but also for the assessment of LTP uninfluenced by anaesthetic agents, which is rare in rodent studies using visual stimulation and is thought to have caused issue with replication in the past [16]. However, in using EEG and the visual LTP paradigm, the rich profile of cortical changes that occur as a result of LTP induction such as those reported in animal invasive studies [3,5,7] are largely lost. Evoked potential changes merely allow for the distinction of relatively increased and relatively decreased LTP or LTD in different groups or conditions in a study.

Computational modelling techniques such as dynamic causal modelling (DCM) and the neural-mass or mean field models typically employed, provide the mathematical means to estimate the pooled response of a large population of neurons [17]. Pertinent to the generation of VEPs, the EEG signal is estimated to include the pooled response of 10,000 to 50,000 neurons [18]. The VEP recorded from the surface of both rodents and humans is known to reflect primarily the superficial layers and the apical projections of pyramidal neurons. However, this response is the product of a hierarchical and reciprocal connectivity structure of excitatory and inhibitory inputs influencing the action of, and acting upon, superficial (II/III) layers; as well as a smaller contribution of the deeper layers [18–21]. This was first described by Douglas, Martin [20] as the canonical microcircuit of cat neocortex. A hierarchical organization of connectivity is described by Felleman and Van Essen [22]. The empirical evidence for basic principles governing connectivity in the brain provides the basis for the development of biologically plausible computational models of the canonical microcircuit that estimate the neural causes of EEG signal.

To better exploit the detail of EEG recordings of VEP enhancement, recent studies in humans have employed computational modelling in network-based analyses of the excitatory cortico-cortical connections involved in visual processing. Connectivity changes associated with the visual LTP paradigm have been assessed using DCM and an adapted, simple Jansen and Rit [23] neural mass model for evoked-potential research. To construct these networks, the sources typically associated with the induced LTP related change include greater activation in the extrastriate visual cortex, dorsal and ventral visual streams, and frontal cortex [13,24,25]. As a result of Hebbian plasticity changes, modulation of excitatory forward projecting connections, as well as local modulation of the cortical microcircuit has been replicated [13,24,25]. However, there is a lack of rodent research on the impact of LTP on cortico-cortical network connectivity to allow for detailed inference on the nature of resulting changes. One way to overcome this would be to model visual cortex more similarly to how it is studied invasively in animals and to assess the reciprocal laminar connectivity changes leading to LTP

triggered VEP modulation. This requires moving beyond the three population, simple Jansen and Rit [23] neural mass model that lacks laminar resolution.

Recently, computational models have expanded to include increasingly detailed recapitulations of the canonical microcircuit and local laminar connectivity. Moreover, conductance models for both spectral and evoked response data capture non-linear as well as linear dynamics intrinsic to a wider range of cell populations and ion channel properties [17]. Particularly pertinent to data containing sensory perturbation, Shaw, Muthukumaraswamy [26] present a model that incorporates realistic thalamic parameters that capture relay projections into layers IV and VI including intrinsic inhibitory and excitatory connections.

The model was employed in the current study with the aim of capturing the induction of LTP in visual cortex following ~9Hz visual stimulation in the classical visual LTP paradigm [4]. It was hypothesised that the changes in laminar connections would be in the same locations as LTP driven changes found by Heynen and Bear [5] and Huang, Rozas [7]. This includes an increase in excitatory connectivity projecting from thalamus into layer IV, and from layer IV into layer III. However, we did not restrict the model to only use these parameters to describe the data. We attempted to balance computational demands with complexity, and allowed all available parameters to vary in layer II/III and IV, as well as all thalamic projections to these layers, reminiscent of the Douglas and Martin [21] model of basic visual processing. This meant less documented and relatively more exploratory changes could be observed, such as inhibitory changes that may potentially be as a result of LTP occurring on inhibitory interneurons. LTP on GABAergic interneurons has been found in hippocampus and cortex [27], including following TBS to visual cortex [28]. LTP on GABAergic interneurons related to visually induced LTP may or may not be NMDA dependent and, as suggested above, such findings will be exploratory.

The specification of a wide range of parameters additionally allows for the model to recapitulate changes that may be inconsistent with LTP or not include any of the hypothesised connectivity changes, for example, decreases in connectivity. The study is therefore a hypothesis driven exploration of visually induced LTP, that could plausibly provide alternative explanations for the modulation of ERPs seen in the visual LTP paradigm.

NDMAR dependent LTP has multiple phases which will be explored using Bayesian Model Selection (BMS) and can be termed short-term potentiation (STP), early phase LTP (E-LTP), and late phase LTP (L-LTP). These phases are dependent on different molecular processes or stages [2,29]. STP is considered relatively unstable, and persists for 15–30 minutes [30]. Therefore, it has been suggested that early potentiation of the VEP (a couple of minutes post-tetanus), that is slightly decayed by the late post-tetanus block (commonly the N1 [4]), is likely contaminated by STP. In contrast, potentiation observed 30 minutes post-tetanus onward is likely to predominantly reflect E-LTP, and may contain some products of the conversion to L-LTP [4,13]. BMS will allow for an investigation of the different time courses of potentiation in VEPs, and how these might be reflective of the interaction of STP as well as E-LTP mechanisms. Specificity of parameter changes to the tetanised sine grating is also modelled. The modelling is accompanied by ERP analyses to confirm the photic tetanus induced significant modulation of the VEP.

## Methods

### Ethics statement

The study was approved by the University of Auckland Human Participants Ethics Committee (UAHPEC reference numbers: 013772 and 8969). Participants provided informed written consent prior to participation.

## Participants

Twenty young adults (sex: 9 males, 11 females), age range: 21–32 (mean = 24.35) took part in the visual LTP paradigm. The data from all females has been previously reported in Sumner, Spriggs [25]. All datasets from females were collected during the follicular phase (day 3–5) of the menstrual cycle when progesterone and oestradiol levels are low in order to control for hormonal effects on LTP [25,31,32]. Participants had no history of neurological or psychiatric disorder and were not taking any psychoactive medications. They had normal or corrected to normal vision.

## EEG acquisition

Continuous EEG was recorded using 64 channel Acticap Ag/AgCl active shielded electrodes and Brain Products MRPlus amplifiers. Data were recorded in Brain Vision Recorder (Brain Products GmbH, Germany) with a 1000 Hz sampling rate and 0.1 μV resolution. As is standard for the ActiCap layout, FCz was used as an online reference and AFz as ground. Electrode impedance below 10 kΩ was achieved prior to recording. Stimuli were displayed on an ASUS VG248QE computer monitor with a screen resolution of 1920 x 1080 and 144 Hz refresh rate. TTL pulses generated through the parallel port of the display computer provided synchronization of stimulus events with EEG acquisition. All stimuli were generated by MATLAB (The MathWorks, Inc., Natick, MA) using the Psychophysics Toolbox [33–35].

## Visual LTP paradigm

Visual LTP was measured using an established paradigm that induces enhancements in the early VEP component as a product of LTP occurring in the visual cortex [4]. The task stimuli were horizontal and vertical sine gratings, with a spatial frequency of 1 cycle per degree. They were presented at full contrast on a grey background, and subtended 8 degrees of visual angle. Participants were seated with their eyes 90 cm from the centre of the screen and were instructed to passively fixate on a centrally presented red dot.

The paradigm comprised four conditions (Fig 1). The first was a pre-tetanus condition; both stimuli were presented in a random order 240 times at 1 Hz for 34.8 ms. The pre-tetanus condition took approximately 8 minutes. The interstimulus interval was varied using 5 intervals from 897–1036 ms and occurred randomly but equally often. The purpose of the pre-tetanus condition was to establish a baseline ERP amplitude for subsequent comparison with post-tetanus conditions. The second condition was a 2 minute photic tetanus or high-frequency stimulation involving 1000 presentations of either the horizontal or vertical stimulus which can be used to test input-specificity [11,12] (counterbalanced between participants) for 34.8 ms with a temporal frequency of approximately 9 Hz. The 9 Hz frequency was chosen as it has been shown to reliably induce potentiation of the VEP [8,11,12,14,36] but is below the rate of perceptual fusion [37]. The interstimulus interval was either 62.6 or 90.4 ms occurring at random but equally often. The third condition, referred to as the early post-tetanus condition, followed a 2-minute break, allowing retinal after images from the photic tetanus to dissipate and to ensure any effects measured were not just attributable to general cortical excitability. The fourth condition, referred to as the late post-tetanus block, followed 35–40 minutes after the early photic tetanus. For both post-tetanus conditions, horizontal and vertical stimuli are presented under the same conditions as the pre-tetanus, but with only 120 presentations of each lasting 4 minutes.

## ERP: Pre-processing

Pre-processing and data analyses were completed using the SPM12 toolbox (http://www.fil. ion.ucl.ac.uk/spm/software/spm12/). Data were re-referenced to the common average. A 0.1–

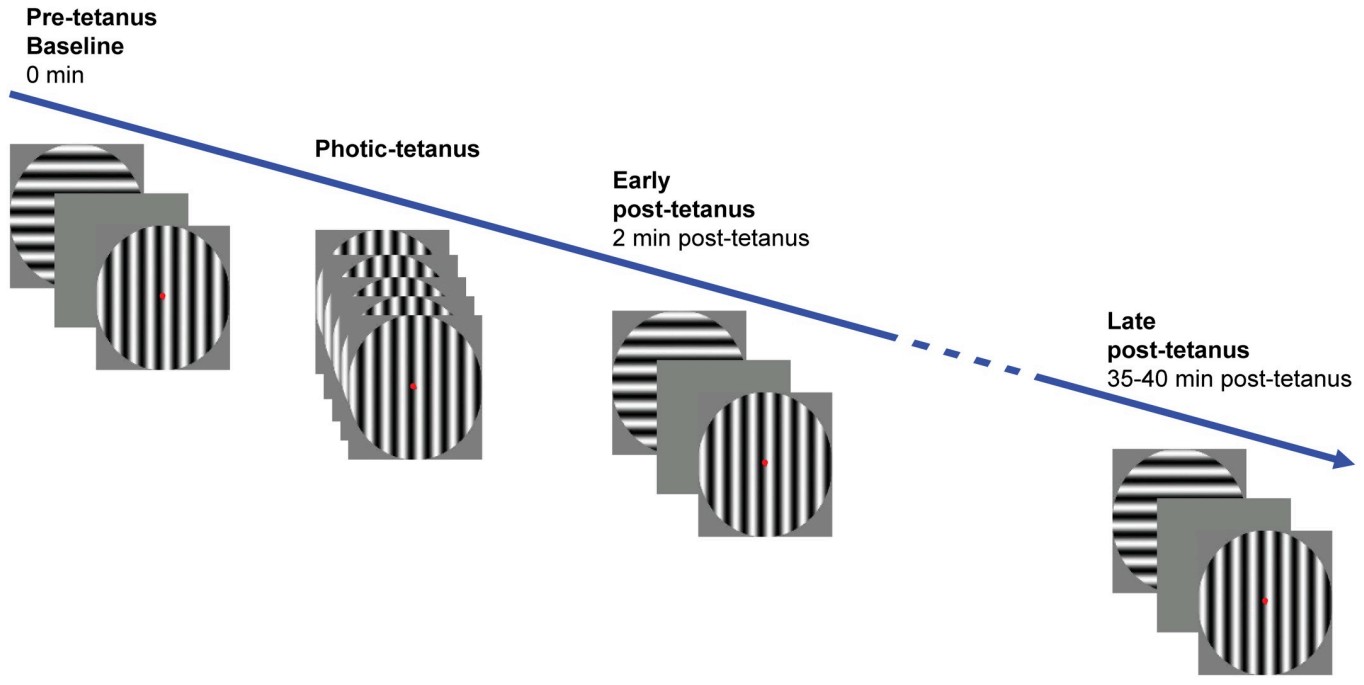

**Fig 1. Schematic of the visual LTP paradigm.**

30 Hz Butterworth bandpass filter removed slow drifts and high frequency noise. The LTP data were epoched into 700 ms sections (-100 ms to 500 ms) and then baselined. Eye blinks were removed using a 100 uV threshold applied to electrodes Fp1, and Fp2 (or AF7, and AF8 if Fp1, and Fp2 electrodes were bad). As the sensory LTP task relies on visual input, and a 100 uV threshold can miss small blinks and saccades along with other muscular and electrical artefacts, manual artefact rejection was also carried out. The manual artefact rejection first consisted of semi-automated trial rejection using the data summary tool from the SPM12 integrated FieldTrip visual artefact rejection toolbox (ft_rejectvisual). Each trial was then visually inspected using the SPM12 integrated FieldTrip data browser (ft_databrowser) to remove any trials containing artefacts that did not meet the threshold of the previous steps. This lead to the removal of on average 34.05 (SD = 28.56) trials from the pre-tetanus condition, 19.50 (SD = 10.75) from the early post-tetanus, and 39.85 (SD = 16.10) from the late post-tetanus conditions, leaving a mean of 445.95, 220.50, and 200.15 trials per condition respectively. Finally, ICA was carried out to remove any residual artefact. Components representing ocular/electrocardiogram artifacts were identified based on their topography and time course and removed for each participant (typically 1 component).

## ERP: Analysis

The pre-processed data were averaged by condition. The trial averages were converted to NIfTI images using a time-window of 0–250 ms. Images were smoothed using a 6×6×6 FWHM Gaussian kernel. ERP analyses in SPM12 represent space by time data as a continuous

statistical parametric map, and employs random field theory to control family-wise error rate. This approach allows for the analysis of large temporal and spatial regions of interest (in the current study this included the entire 0–250 ms time-window and 19 electrodes) while also controlling for the multiple comparisons problem [38].

The purpose of the ERP analysis was to confirm and demonstrate evidence of VEP modulation post-tetanus. As in previous studies an occipito-parietal ROI was employed, comprising the following electrodes: P1, P2, P3, P4, P5, P6, P7, P8, Pz, PO3, PO4, PO7, PO8, PO9, PO10, POz, O1, O2, and Oz [24,25]. A 3-way repeated measures ANOVA (pre tetanus x early post-tetanus x late post-tetanus) was implemented across the 250 ms time window. For this analysis the contrasts of interest were an effect of the high frequency stimulation on both the early post-tetanus and late post-tetanus blocks. Only the stimulus used in the photic tetanus for individual participants (horizontal or vertical sine gratings depending on allocation) were used in the ERP analysis for simplicity. Simple effects post-hoc analyses were carried out as appropriate to explore main effects and interactions. Results were interpreted at the peak level, and effects were considered significant at $p < 0.05$ family-wise error corrected (FWE-c).

## Source analysis

Sources analysis was carried out using Multiple Sparse Priors, as implemented in SPM12, for group inverse reconstruction [39]. As individual MRI structural scans were not available, the template head model was used (based on the MNI brain) with the 'normal' cortical mesh. For the forward computation the standard head model option for EEG is the Boundary Element Method (BEM). To define electrode locations a standard template for the 64 channel ActiCap, including standardised fiducials, was used. The source analysis was run on the time-window 100–220 ms to balance encompassing the major effects of the photic tetanus on the evoked response determined from the ERP analysis, with a narrow time-window to limit the number of multiple comparisons. A 3-way repeated measures ANOVA was run (pre tetanus x early post-tetanus x late post-tetanus). A peak was selected using the FWE-c F contrast [1–1–1] that encompassed both early and late effects of the photic tetanus, allowing a single node to be modelled. The selected peak was extracted from each participant as the local field potential (LFP), with a 5 mm spherical radius around the MNI coordinate using spm_eeg_inv_extract. Projecting the data into the brain as a local field potential (LFP) allows for the solving of the spatial forward model prior to solving the observation model, this is computationally more expedient for example by removing tissue conduction as a potential source of variation at the observation model fitting stage.

## Computational model of microcircuitry

This study implemented a thalamocortical model of interlaminar connectivity (Fig 2). The model builds upon and extends the cortical 'cmm_NDMA' model implemented in DCM [40–42], employing Morris-Lecar conductance equations [43], and summarised as a neural-mass [44].

The 'cmm_NMDA' model features 4 interconnected populations comprising layer IV spiny stellates, superficial (layer II/III) and deep (layer V) pyramidal populations, and a single pool of inhibitory interneurons. The thalamocortical model presented here extends this architecture to include an additional set of cortical inhibitory interneurons (allowing separate superficial and deep populations). Furthermore, we include an additional population of cortico-thalamic (aka thalamic projection) pyramidal neurons in layer VI [45]. For the thalamus, we model one population of (excitatory) relay cells and one population of (inhibitory) reticular cells.

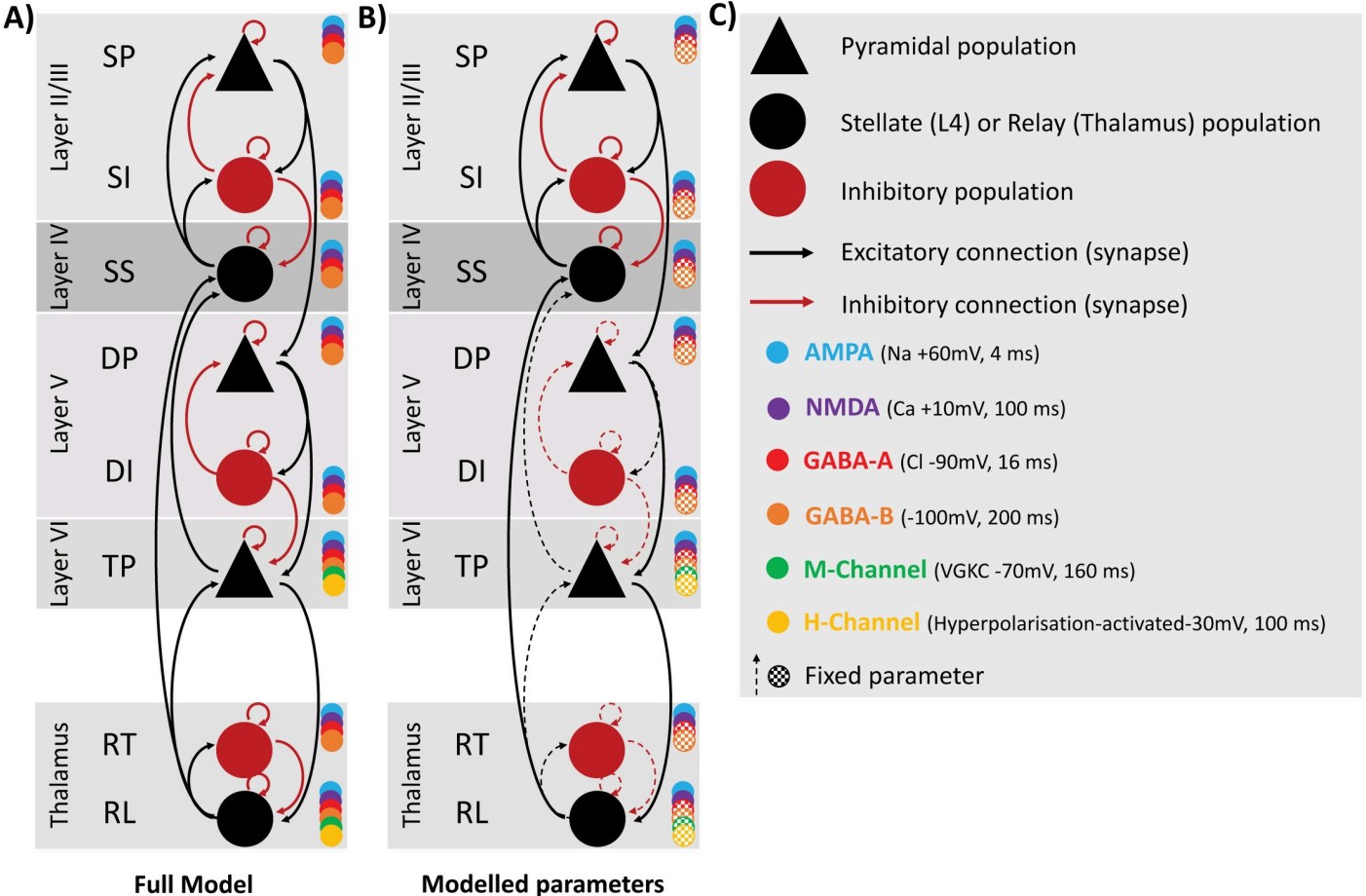

**Fig 2.** A) Thalamocortical model architecture and connectivity with 6 cell populations in the cortical column, and 2 thalamic populations. Cortical populations include layer II/II with superficial pyramidal (sp), and superficial inhibitory interneuron input (si). Layer IV, the granular layer, has spiny stellate input, and is also the source of thalamic relay input as well as sp input. Layer V is populated by deep pyramidal (DP), and deep inhibitory interneuron input (DP). Layer VI is populated with the thalamic projection (TP) cells, the source of relay input also. The model is also furnished with the decay constant of AMPA, NMDA, GABA-A, GABA-B, and M- and H channels. B) The parameters are depicted in terms of whether their priors were fixed or allowed to vary for model inversion. Parameters that were allowed to vary included those outlined by Douglas and Martin [21] as being key for visual information processing, and the corresponding within-layer connections in II/III and IV C) Key for A and B.

All populations are governed by conductance equations of the form:

$$C\frac{dV}{dt} = \sum g_n(V - V_n) + u$$

$$\dot{g}_n = \kappa_n(\varsigma_n - g_n)$$

$$\varsigma_n = \gamma_{i,j}\sigma(\mu_v^j - V_R, \sum j)$$

$$n = \{AMPA, NMDA, GABA_A, GABA_B, m, h\} \tag{1}$$

Here, the mean membrane potential (dV/dt) of a population is represented by the sum of the different channels' ($n$) conductance terms ($g$), multiplied by their respective reversal potential ($V_n$), plus any exogenous / endogenous input ($u$).

The rate of change of conductance of channel $n$, denoted $g_n$, is calculated from the rate constant of the channel ($K_n$) multiplied by the change in conductance, e.g. $\kappa_n(\varsigma_n - g_n)$. The conductance, $\varsigma_n$, is calculated from the third term in equation 1, where the coupling parameter, $y_{ij}$, coupling population $j$ to $i$, is multiplied by the expected firing of source population $j$. The sigmoid function sigma represents the cumulative distribution function of the presynaptic depolarisation around a threshold potential of $V_R = -40$ mV, which determines the proportion of cells firing.

Conductances were computed for 6 channel types; AMPA, NMDA, GABA$_A$, GABA$_B$, m-currents and h-currents. M- and h- channels were only present on cortico-thalamic projection pyramids (layer 6) and on thalamic relay cells.

The full connectivity profile of the model is depicted in Fig 2. Excitatory connections (black in Fig 2A) are modelled by AMPA and NMDA conductances whereas inhibitory connections (red in Fig 2A) are modelled by GABA$_A$ and GABA$_B$ conductances. As per previous formulations [40], NMDA channels are subject to voltage gating to represent magnesium block, represented by a sigmoid activation function:

$$f_{MG}(V) = \frac{1}{1 + 0.2\exp(-\alpha_{NMDA}V)} \qquad (2)$$

Such that the conductance-based equation for layers where NMDA is an included population becomes:

$$C\frac{dV}{dt} = \sum g_n(V_n - V) + (g_{NMDA} * f_{MG}(V) * V_{NMDA} - V)) + u \qquad (3)$$

The rate constants and reversal potentials of each channel are represented in Fig 2C. Thalamo-cortical projections were subject to propagation delays of 3 ms and cortico-thalamic of 8 ms [46]. The parameterisation of the neural and observation models for inversion to the empirical ERPs can be found in the S1 Text.

For this analysis we constrained the parameters that were allowed to vary (during model fitting) according to the Douglas and Martin [47] canonical microcircuit. The current model is more detailed than the Douglas and Martin [47] canonical microcircuit so we also allowed all of the additional parameters within layers II/III and IV to vary. In summary: rl ➔ ss, ss ➔ss, ss ➔ sp, ss ➔ si, si ➔ sp, si ➔ si, sp ➔ sp, sp ➔ si, si ➔ ss, sp ➔ dp, dp ➔ tp, tp ➔ rl, were allowed to vary and the rest of the parameters were fixed (dotted lines in Fig 2B). As we hypothesised that there would be specific effects on AMPA, and potentially NMDA, the decay rates of these channels were also allowed to vary in the model; GABA$_A$ and GABA$_B$ decay rates were fixed.

The generative model described above was fit to the empirical EEG data using DCM. Since DCM permits simultaneous inversion of multiple experimental conditions, by incorporating a parameterised general-linear model (GLM) into the inversion protocol, we compared three plausible models. First, LTP was modelled as a linear change from baseline that is greatest in the late-post tetanus block [–1 0 1], to reflect early phase LTP and the typical time course of P2 potentiation [4] (Fig 3). As the first measure of potentiation takes place 2 minutes post-tetanus it would be expected that there would be contamination of STP in the first post-tetanus block but not the second. It has been reported that this may explain why potentiation of the N1 begins to degrade by the late post-tetanus block [4,13]. For this, a non-linear change from baseline, that peaks in the first post-tetanus block (to model contamination with STP), was modelled [–1 1 0] (Fig 3). Additionally, the contrast [-1 1 0; -1 0 1] allowed for both non-linear and linear contributions to describe the condition specific effects (Fig 3). The winning model was determined using fixed-effects (FFX) Bayesian Model Selection (BMS). FFX was

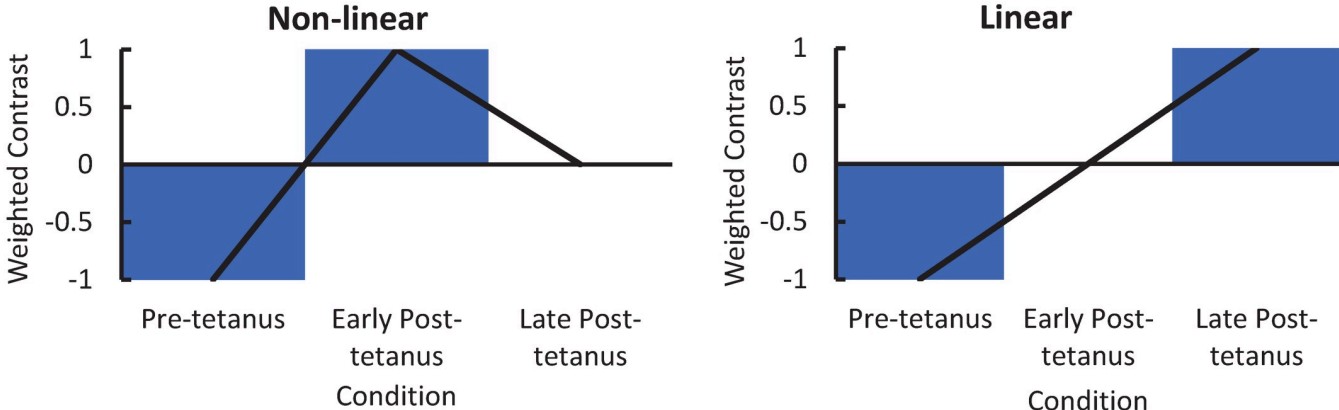

**Fig 3. Contrast designs for the non-linear [–1 1 0] and linear [–1 0 1] models, where the combination is modelled as having a non-linear and linear contribution [-1 1 0; -1 0 1].**

determined to be appropriate given that the biological underpinning of LTP can be assumed to be the same for all healthy humans [48]. Random-effects (RFX) results are provided additionally to FFX as in Shaw, Hughes [49]; RFX gives Bayesian Omnibus Risk (BOR)—the log-Bayes factor which provides the probability of a model's frequency providing evidence for H1 over H0. A BOR of ≤ 0.25 is taken as strong evidence in favour of the winning model (Rigoux et al., 2014). Protected exceedance probabilities (pxp) provide a probability of the most frequent model, over all other tested models (Rigoux et al., 2014).

Following BMS, a RM-MANOVA that analysed the effect of condition (pre-tetanus, early post-tetanus, and late post-tetanus) on the change in interpopulation synaptic coupling, and time constants was conducted for the linear and non-linear models. Univariate ANOVAs were used to determine the parameters that contributed to a significant effect of condition; corrected for multiple comparisons using false discovery rate (FDR) [50]. Statistics on the parameter estimates were carried out using SPSS 25 (Armonk, NY).

The primary outcome was the model output for the tetanised data, however, the model was also run on the non-tetanised data to explore if the thalamo-cortical model demonstrates evidence of specificity of parameter changes to the orientation of the tetanised grating.

Next, due to the relative novelty of parameterising and modelling thalamic connectivity, the additional step of using BMS to compare the winning model with the same model but with no thalamic compartment is provided, replacing thalamic input with a gaussian bump function into layer 4. BMS has already determined that thalamo-cortical model is superior to simpler alternatives such as the 4 population CMC model this model was built on (cmm_NMDA) [26].

## Results

### ERP: Analysis

The purpose of the ERP analysis was to confirm that the photic tetanus had induced typically seen modulation of the visually evoked potential. There was a significant effect of photic tetanus on both post-tetanus blocks (using the F contrast [1–1–1]) including a central peak at 224 ms ($F_{(1,57)}$ = 136.34, $p$ = 7.58e$^{-13}$) and central ($F_{(1,57)}$ = 132.92, $p$ = 1.22e$^{-12}$) and right lateralised

($F_{(1,57)}$ = 128.27, $p$ = 2.36e$^{-12}$) significant peaks at 196 ms within the same cluster. An early centralised peak was also significant at 80 ms ($F_{(1,57)}$ = 41.30, $p$ = 4.45e$^{-5}$), with a significant right lateralised peak at 128 ms ($F_{(1,57)}$ = 29.29, $p$ = 0.0013), and a 132 ms ($F_{(1,57)}$ = 27.34, $p$ = 0.0023) peak in the cluster. Finally an additional early central peak also occurred at 76 ms ($F_{(1,57)}$ = 21.20, $p$ = 0.015). Despite the relatively conservative nature of FWE-c, a significant peak also occurred at 8 ms ($F_{(1,57)}$ = 34.74, $p$ = 0.0003), as it is too early to represent visual processing, it likely represents noise.

The *post-hoc* contrast using the *t*-contrast [1–1–1] revealed that the evoked response was significantly more negative in the peaks at 80 ($t_{(57)}$ = 6.43, $p$ = 2.22e$^{-5}$), 128 ($t_{(57)}$ = 5.41, $p$ = 0.001) and 132 ms ($t_{(57)}$ = 5.41, $p$ = 0.001). These findings are consistent with changes to the N1 component. Using the *t*-contrast [–1 1 1] the evoked response was more positive in the peaks at 224 ($t_{(57)}$ = 11.68, $p$ < 6.12e$^{-13}$), 196 ms ($t_{(57)}$ = 11.53, $p$ < 6.12e$^{-13}$); ($t_{(57)}$ = 11.33, $p$ < 1.18e$^{-12}$) and 76 ms ($t_{(57)}$ = 4.60, $p$ = 0.008). The 196 to 224 ms peaks appear to represent and the P2 component becoming more positive, and at 76 may reflect the P1 becoming more positive. The P2 increase in the post-tetanus block is illustrated in Fig 4, and also appears to capture a significant decrease in amplitude around the 128–132 ms.

Exploring the early and late post-tetanus conditions separately, the *t*-contrasts [1–1 0] showed the significant peak at 132 ms reported above ($t_{(57)}$ = 3.99, $p$ = 0.043) and [–1 0 1] showed the significant peak at 192 ms ($t_{(57)}$ = 4.31, $p$ = 0.018), consistent with the tendency for the N1 to be most significant in the early post-tetanus block, and the P2 in the late [4]. The alternate direction contrasts for each condition had no significant peaks. The early effects are evident on the leading edge of the P2 at POz, in addition to the late effect on the peak of the P2 (Fig 4). A summarising image of the results using the maximum intensity projections can be found in S1 Fig.

## Modelling: Analysis

Source analysis revealed a significant effect of photic tetanus that was particularly strong in visual cortex. As the main peaks were in calcarine fissure and yet the effect of visual LTP has been most typically studied in extrastriate cortex [8,13,24,25,51], a peak in left middle occipital gyrus (MOG) was selected from the literature [24] that was still within the significant area of activation for the current study: MNI [–36, –90, 4]. This peak was extracted as a local field potential for all participants and described a mean variance of 77% (SD = 17%) of the scalp data (averaged across a cluster including and around POz comprising P1, Pz, P2, PO3, P4).

## Bayesian model selection on the thalamo-cortical model for tetanised data

FFX revealed that the combination model was the winning computational model (Fig 5). RFX BOR = 0.086 provides very strong evidence for the combination model (pxp = 0.49), closely followed by the linear model (pxp = 0.46), and ~16-fold weaker non-linear model (pxp = 0.03). While the result could be considered a split model win for the linear and combination model, the combination model does include the linear model, and extra correction is applied to the combination model because it contains extra degrees of freedom introduced by extra free parameters.

The individual data fits explained >80% variance for all participants for all models. The modelled response in Fig 4 clearly recapitulates the condition effects seen in both the surface recorded ERP at electrode POz and the source reconstructed response.

Because we were interested in establishing the parameters that contributed to the short-term and long-term potentiation, and the individual fits for all models were good, we extracted the values for the non-linear and linear models separately.

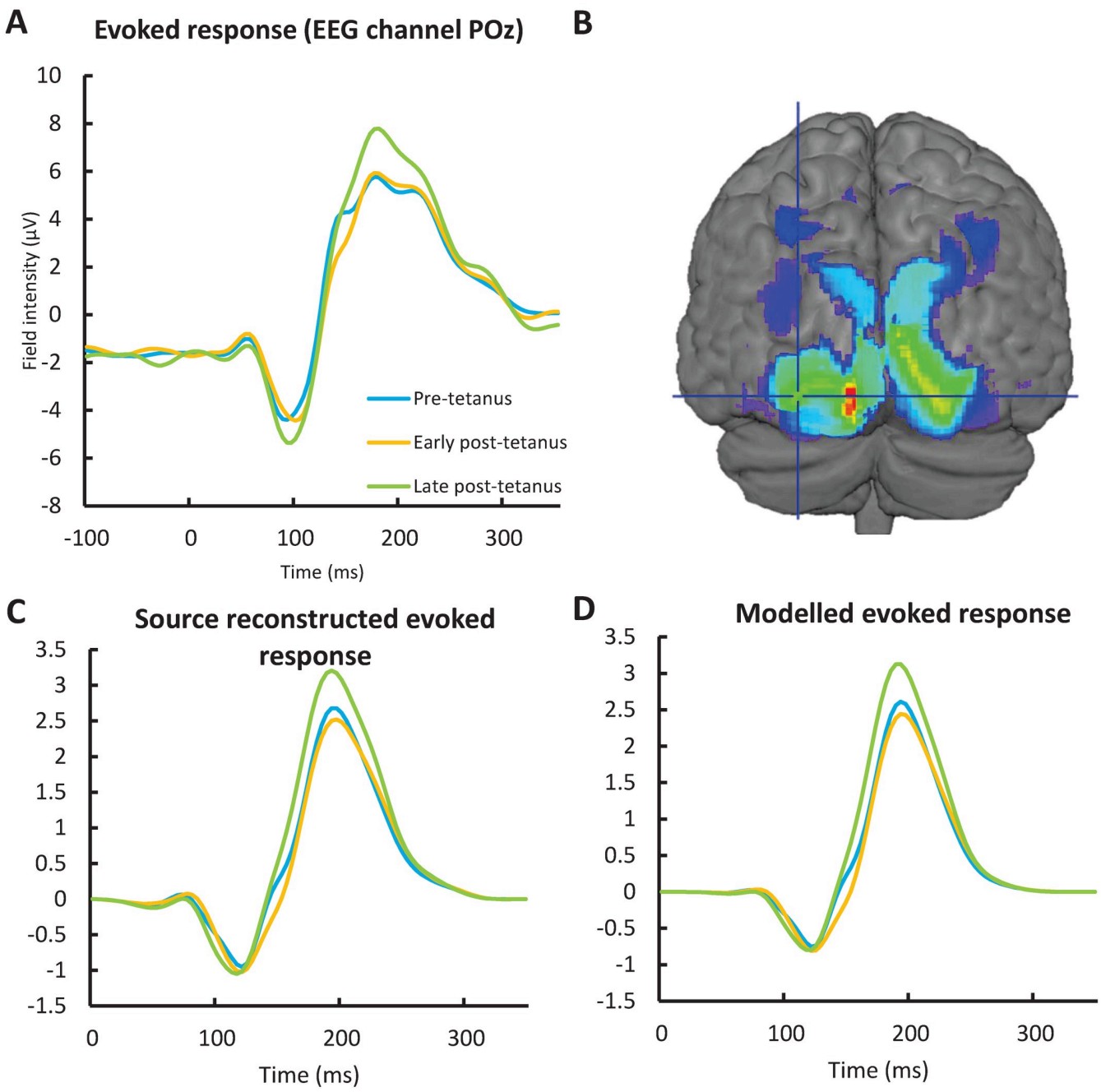

**Fig 4.** A) The VEP as recorded at electrode POz shows evidence of the differences pre-tetanus compared to both early post-tetanus and late post-tetanus found in the peak field effects. This includes (least clearly shown) a peak at 132 ms that was most significant in the early post-tetanus block, and (most clearly shown) at around 196 ms a peak that was most significant in the late post-tetanus block. B) Results of the source analysis looking at pre- vs early and late post-tetanus. Thresholded to p<0.05 uncorrected for illustration purposes. The crosshair lies over the literature selected peak coordinate [-36. -90, 4] in left middle occipital gyrus. C) The time related effects post-tetanus are recapitulated in both the source reconstructed evoked response, D) as well as the evoked response produced at the source by the model. However, while expected, some loss of information from the scalp recorded ERP is apparent. The shift in latency of the early negativity is considered a more complete representation of the (usually more bilateral) N1 than shown at POz.

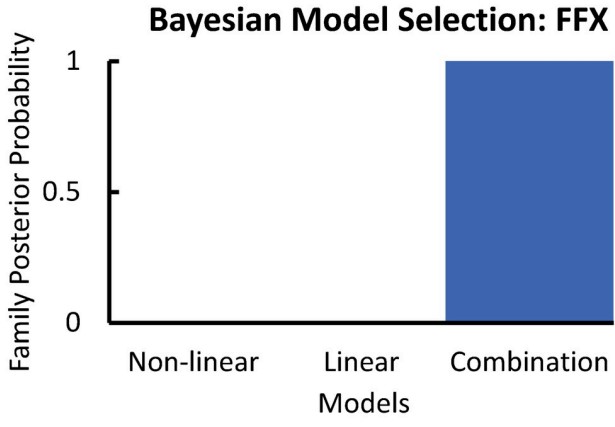

**Fig 5. Results of the BMS using FFX demonstrating that a combination of both linear and non-linear effects best describe the data.** The next best model represented a difference in log-evidence (ΔF) between the two highest scoring models of 1382.86 for the linear model, and 1554.79 for the combination model, relative to the non-linear model alone. This indicates >99% posterior confidence in the winning model.

The RM-MANOVA for the effect of condition (pre-tetanus, early post-tetanus, and late post-tetanus) on the interpopulation synaptic coupling, and time constants that was conducted for the non-linear model revealed a significant effect of condition ($F_{(26,54)} = 7.045$, $p = 1.514e^{-9}$) using Wilk's lambda $\lambda = 0.059$ (Fig 6). The univariate ANVOA results (Fig 6) revealed that this was driven by a significant increase in conductance in the connection ss➜sp ($F_{(2,54)} = 22.36$, $p = 4.203e^{-6}$ FDR), and approaching significant increase in ss➜si ($F_{(2,54)} = 4.429$, $p = 0.057$ FDR, or 0.019 uncorrected). There was a significant decrease in sp➜dp ($F_{(2,54)} = 14.641$, $p = 0.0001$ FDR). The time constant for AMPA was significantly increased ($F_{(2,54)} = 11.139$, $p = 0.0006$ FDR).

The RM-MANOVA for the linear model revealed a significant effect of condition ($F_{(26,54)} = 6.404$, $p = 8.120e^{-9}$) using Wilk's lambda $\lambda = 0.068$ (Fig 6). The univariate ANOVA results (Fig 6) revealed a significant increase in ss➜sp ($F_{(2,54)} = 6.079$, $p = 0.015$ FDR), sp➜sp ($F_{(2,54)} = 6.536$, $p = 0.015$ FDR), si➜ss ($F_{(2,54)} = 18.902$, $p = 0.000002$ FDR), and rl➜ss ($F_{(2,54)} = 13.573$, $p = 0.002$ FDR).

## Bayesian model selection on the thalamo-cortical model vs. a cortex-only model

For the winning, combination linear-nonlinear model, we explored the possibility that a cortex-only version of the model (i.e. without the thalamic populations) would suffice. Using BMS, we compared the thalamic model to a cortex only model using BMS, which clearly showed very strong evidence for the model with thalamic cells (FFX = 1; RFX-BOR = $0.1e^{-4}$; pxp with thalamus = 0.9565, without = 0.0435). S2 Fig provides the relative log-evidence for each model.

## Bayesian model selection on the thalamo-cortical model for non-tetanised data

The equivalent analysis run on the non-tetanised data produced individual data fits that explained >90% variance for all participants for all models. BMS comparing the nonlinear, linear and combination models, for the non-tetanised data, also demonstrated preference for the

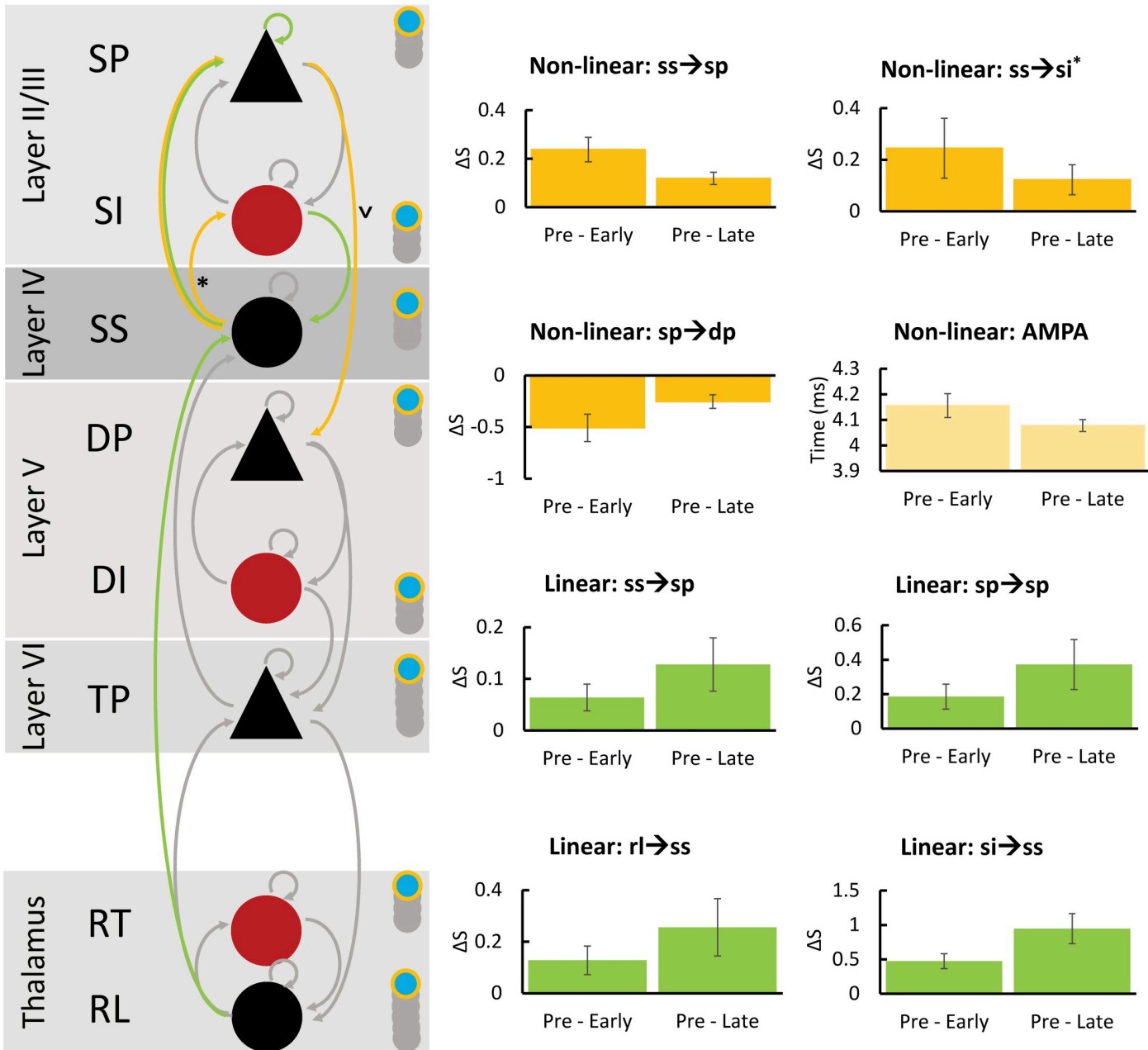

**Fig 6. The RM-MANOVA on the task-specific effects for both the non-linear (yellow) and linear (green) models demonstrated that the connectivity between spiny stellates (ss) and superficial pyramidal cells (sp) were greater than pre-tetanus in both the early and late post-tetanus conditions.** Non-linear task-specific effects included significant changes in ss to superficial inhibitory interneuron (si) connectivity (* indicates uncorrected), and sp to deep pyramidal (dp) connectivity. For sp to dp this was a decrease in connectivity (connection indicated by a ˅). The AMPA decay constant was significantly longer than pre-tetanus for both conditions post-tetanus, but degraded by late post-tetanus. For linear effects, the self-gain of sp was significant, as was si to ss. Thalamic relay (rl) to ss was also significant.

combination model (FFX = 1, RFX pxp = 0.5063, see S3 Fig). As with the tetanised model there is evidence for split model win using RFX for the linear model pxp = 0.41, and non-linear was much weaker pxp = 0.09.

An identical RM-MANOVA was run on the linear and non-linear model separately. The RM-MANOVA for the non-linear model revealed a significant effect of condition ($F_{(26,54)}$ = 2.898, $p$ = 0.0001) using Wilk's lambda $\lambda$ = 0.167 (Fig 7). The univariate ANOVA result revealed a significant increase in ss➔ss ($F_{(2,54)}$ = 5.878, $p$ = 0.0197 FDR), ss➔si ($F_{(2,54)}$ = 6.772, $p$ = 0.013 FDR), and AMPA ($F_{(2,54)}$ = 7.537, $p$ = 0.013 FDR) (Fig 7). There was a significant decrease in sp➔sp ($F_{(2,54)}$ = 7.547, $p$ = 0.013 FDR).

The RM-MANOVA for the linear model revealed a significant effect of condition ($F_{(26,54)}$ = 3.957, $p$ = 0.000012) using Wilk's lambda $\lambda$ = 0.113 (Fig 7). The univariate ANOVA result revealed a significant increase in si➔ss ($F_{(2,54)}$ = 13.026, $p$ = 0.00007 FDR), and rl➔ss ($F_{(2,54)}$ = 7.247, $p$ = 0.014 FDR).

## Discussion

The current study set out to validate a thalamocortical model of the microcircuit structure of visual cortex for exploring event related potential data, in particular to explore its utility in the non-invasive visually induced LTP paradigm. It was hypothesised that the model would demonstrate connectivity changes in the same locations as found in invasive animal research, providing an *in vivo* method for measuring changes to the laminar connectivity in visual cortex induced by LTP. The use of non-linear and linear contrasts was set up to capture the STP as well as the E-LTP driven parameter changes. The combination of both was found to best describe the data, providing validation for the idea that the trial specific changes seen in the ERPs contain both STP and E-LTP effects on the underlying cortical microcircuitry [4,13]. The visual LTP paradigm was found to induce both linear and non-linear changes to the excitatory connection from ss to sp, indicating the contribution of both STP and LTP mechanisms. A linear increase in rl (thalamic) to ss (layer IV) excitatory connectivity also indicated LTP related increases had occurred. Both changes had been demonstrated by Heynen and Bear [5] following photic and TBS induction protocols in rodents ss (layer IV) to sp (layer II/III) connectivity has also been shown using TBS induction in primates by Huang, Rozas [7]. Additional changes were found, potentially due to the nature of the larger field recorded from the scalp with EEG compared to layer specific implanted electrodes. The changes included ss ➔ si, and si ➔ si reciprocal connectivity, sp self-gain, and a decrease in the sp ➔ dp connectivity. The decay constant of AMPA was also increased from pre-tetanus. Overall, the model provides remarkably plausible changes to the microcircuit following visually induced LTP.

The visual LTP paradigm benefits from the fact that the visual cortex is a primary site of LTP research in neocortex. This led to the ability to make strong hypotheses in the current study of how the thalamocortical model would recapitulate the effects of photic tetanus in laminar connectivity changes. As briefly introduced above, Heynen and Bear [5] have demonstrated that potentiation leads to excitatory feed forward connectivity into layer IV rising from modification of the thalamocortical projection, evidenced also in the modification of field potentials [5]. They also showed contemporaneous increases in superficial layer II/III current sinks. This was established as being a separate site of LTP [5]; the finding was further supported by Huang, Rozas [7] who demonstrated inducing LTP at layer IV potentiated field potentials at layer III. Our results produced connectivity changes between feedforward connections from layer IV to II/III, providing encouraging support for the likelihood that the model was producing plausible effects consistent with connectivity changes seen in invasive animal LTP research. The effects on the model also provide evidence that the visual LTP paradigm is producing changes to laminar connectivity consistent with LTP driven effects, as well as modulation of VEPs post-photic tetanus. Further, the consistency of the modelled results with changes seen in electrical induction protocols indicates that the photic tetanus may be

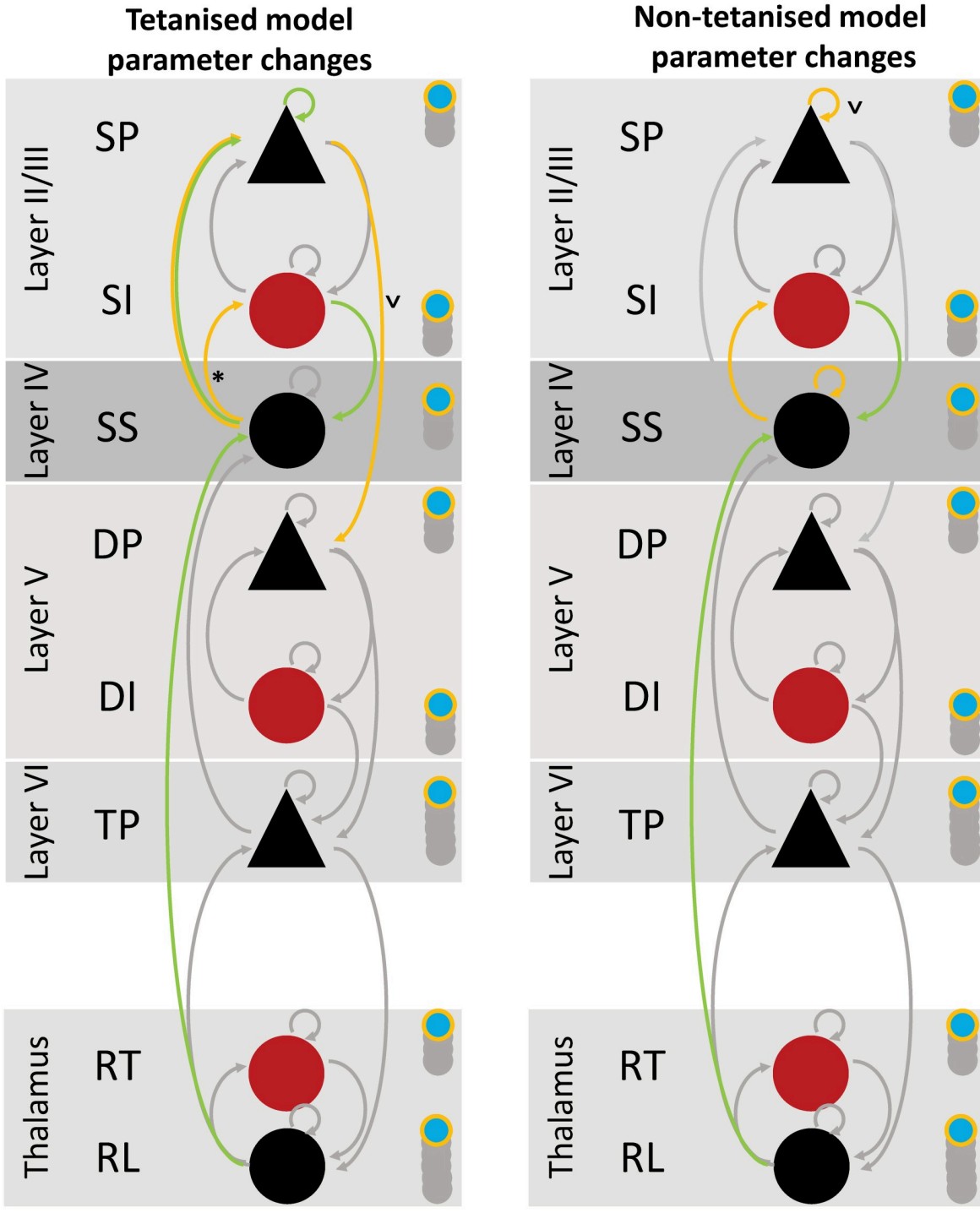

**Fig 7. Difference in significant parameter changes for the tetanised compared to non-tetanised model.** Note. Tetanised model results are identical to depiction in Fig 6 and are provided here for side-by-side comparison. See Fig 2C for key of parameters.

activating neural circuitry in a manner similar to electrical tetanic bursts as hypothesised in Clapp, Hamm [9]. Future steps to validate the model may include adapting and applying it to invasive animal research on LTP in the visual system. This could be done using data such as that already acquired in Heynen and Bear [5] or [14], similarly to the use of DCM in rodent LFP recordings in Moran, Jones [41].

In addition to the changes to thalamocortical connectivity reported above, further changes found may reveal additional insight into the effect of the photic tetanus on laminar connectivity, though these findings are relatively more exploratory. There was a significant non-linear decrease in the excitatory connection from sp (layer II/III) ➔ dp (layer V) that may be explainable by LTD. LTD is as important to learning and memory as LTP [52]. Homosynaptic LTD is a long-term decrease or weakening of synaptic efficacy [52]. In contrast to LTP, LTD is induced by slow ~1-3Hz stimulation [53–55]. It has previously been speculated that the ~1 Hz or slow stimulation in the visual LTP paradigm may induce homosynaptic LTD related changes to the VEP, in addition to LTP from the photic-tetanus [36,56,57]. However, the non-linear change in the connection from sp ➔ dp indicates the contribution of a shorter-term depression that does not return fully to baseline by the late post-tetanus block. This may be consistent with reports by Forsyth, Bachman [15] on heterosynaptic LTD occurring with the visual LTP paradigm. The limited research on heterosynaptic LTD in neocortex demonstrates relatively greater transience (~10 min) than homosynaptic LTP and LTD [58]. Heterosynaptic LTD provides a number of important mechanisms for maintaining stable neuronal networks, by preventing runaway dynamics inherent in a purely Hebbian plasticity system governed by homosynaptic processes [59]).

While a transient activation of NMDARs initiate LTP, it is expressed as a persistent increase in AMPAR transmission [60]. Relatively greater changes to AMPARs including the phosphorylation and trafficking of new receptors occur in the first 40 minutes and have been shown to change AMPAR time dynamics and opening probability [60–64]. The AMPAR numbers begin to decline by 4 hours [61,63]. While changes to the decay constant for AMPARs or NMDARs has not typically been found in invasive research, when measured as pooled changes, the increase in variance caused by the relatively larger number of receptors and higher opening probability may plausibly lead to increases in the mean decay constant, as found in the current study. There is a slower increase in the trafficking of NMDARs, that restores the ratio of AMPAR to NMDAR signalling within 2 hours [63], and may be why this effect was only seen in the AMPAR time constant and not NMDAR. The thalamocortical model does not calculate single channel conductance that would have better allowed for quantification of the change in AMPARs.

The role of inhibitory interneurons in Hebbian long-term potentiation is less well established. However, LTP on inhibitory interneurons has been demonstrated in both visual cortex and hippocampus [27,28], it may not always be Hebbian, and it may or may not be NMDAR dependent. The finding of increased inhibitory input into ss, with the reciprocal but shorter-term increase in ss ➔ si input is a plausible and interesting interaction of LTP. In rodents, layer IV somatosensory cortex has shown that thalamocortical input mediates LTP in inhibitory interneurons [65]. Feed-forward inhibition into layer IV truncates LTP in ss cells and is essential for specific, coordinated, and healthy sensory processing [65]. It is not clear whether the change observed in the current study is NMDA dependent and provides an interesting area for future exploration in clinical and pharmacological research.

Modelling was also carried out on the non-tetanised data and demonstrated evidence that the thalamo-cortical model is sensitive to the orientation of the tetanising stimulus. In the visual LTP ERP literature, the N1 has demonstrated input specificity to the spatial resolution and orientation of the tetanised grating [11–13]. Specificity of LTP to the spatial resolution of

sine gratings has also been shown in rodents [3]. The P2 does not demonstrate input specificity to these stimulus properties and it has been posited P2 may be sensitive to some other undetermined stimulus characteristic, as P2 is generated later in the visual processing pathway than N1 [4]. Therefore, in the current study modelling the non-tetanised data was expected to cause broadly similar changes to the tetanised data, however, representing the contribution of orientation specific LTP to the evoked response data, less parameters would be expected to be significant. This is exactly what was found (Fig 7). Half as many linear parameters were significantly increased (si ➜ ss, and rl ➜ ss). Half as many of the same parameters as in the tetanised non-linear model were significantly increased for the non-tetanised non-linear model (ss ➜ si, and AMPA). Notably absent from the non-tetanised stimulus compared to the tetanised stimulus was significant modulation of the connection from layer IV to III via ss ➜ sp. Additionally, the sp ➜ sp connection moved from a significant linear increase in the tetanised only model, to a non-linear decrease. This is speculated to represent concurrent transient LTD in superficial layers in the non-tetanised data. The additional self-gain parameter ss ➜ ss was significantly increased for the non-linear model.

Overall, these data from the non-tetanised model output provide evidence that the presented thalamo-cortical model is sensitive to the orientation of the tetanising input grating. In particular the increase in rl ➜ ss, and also AMPA parameters indicate that LTP still occurs in the non-tetanised data, although to a lesser extent (in fewer populations of cells). Evidence of connectivity and AMPA changes consistent with LTP in the non-tetanised model is interpreted as the contribution of cortical microcircuitry involved in later visual processing. This interpretation could be tested in future research using hemispheric input specificity via ocular occlusion, whereby no significant LTP driven parameter modulation would be expected to occur in visual cortex of the non-tetanised hemisphere (based on ocular dominance and LTP reviewed by Cooke and Bear [66] and visual LTP results from Teyler, Hamm [8]).

The changes to the visual laminar connectivity in the thalamocortical model following visually induced LTP demonstrated remarkable consistency with the results of rodent invasive research. In particular the increase in conductance in the excitatory connection from the thalamic relay pathway to layer IV, and from layer IV to layer II/III, established major sites of LTP in visual cortex [5,7,67]. Additional changes include LTP at inhibitory interneurons and the possibility of LTD decreasing excitation in the projection from layer II/III into layer V. These findings provide justification for the implementation of the presented thalamocortical model for ERP research in order to provide increased detail on the nature of changes that underlie LTP induced in visual cortex, and for translating rodent finding of the deficits to LTP that may underlie neurological and psychiatric disease to non-invasive research in humans.

## Supporting information

**S1 Fig. Event related potential maximum intensity projections.**
(DOCX)

**S2 Fig. Bayesian model selection for the thalamo-cortical and cortical-only model.**
(DOCX)

**S3 Fig. Bayesian model selection for the non-tetanised data.**
(DOCX)

**S1 Text. Parameterisation of the neural and observation models and supporting results.**
(DOCX)

## Acknowledgments

We would like to thank Professor Ian Kirk and Associate Professor Suresh Muthukumaraswamy for the use of data.

## Author Contributions

**Conceptualization:** Rachael L. Sumner.

**Formal analysis:** Rachael L. Sumner, Meg J. Spriggs, Alexander D. Shaw.

**Methodology:** Rachael L. Sumner, Alexander D. Shaw.

**Writing – original draft:** Rachael L. Sumner, Alexander D. Shaw.

**Writing – review & editing:** Rachael L. Sumner, Meg J. Spriggs, Alexander D. Shaw.

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
