## [Decision Letter · Decision Letter 0]

6 Jul 2020

Dear Dr Sumner,

Thank you very much for submitting your manuscript "Modelling thalamocortical circuitry shows visually induced LTP changes laminar connectivity in human visual cortex" for consideration at PLOS Computational Biology.

As with all papers reviewed by the journal, your manuscript was reviewed by members of the editorial board and by several independent reviewers. In light of the reviews (below this email), we would like to invite the resubmission of a significantly-revised version that takes into account the reviewers' comments.

We cannot make any decision about publication until we have seen the revised manuscript and your response to the reviewers' comments. Your revised manuscript is also likely to be sent to reviewers for further evaluation.

Sincerely,

Hugues Berry

Associate Editor

PLOS Computational Biology

Daniele Marinazzo

Deputy Editor

PLOS Computational Biology

Reviewer's Responses to Questions

**Comments to the Authors:**

Reviewer #1: This paper describes the use of a thalamo-cortical DCM to determine the locii of LTP in a visual stimulation EEG paradigm. I greatly enjoyed reading this paper - I found the idea really interesting, and the design and analysis well executed. My main concern is that the model is not sufficiently probed using control analyses. However, if this concern was addressed, I would have no problem recommmending this paper for publication.

Major comments

More details should be provided about the source analysis. How many spatial, temporal modes were used? What was the sensor level covariance matrix? What forward model was used? I'm guessing that template surfaces and/or MRI volumes were used for the forward model. How were the fiducial coordinates defined?

The main piece that I think is missing is any sort of control condition and model. Most crucially, what about post-tetanus VEPs to the grating orientation not used for the tetanus? Would the model correctly identify no changes in its parameters? I think it's fine if you don't want to include this in the ERP analysis (making it a 4-way ANOVA). I think you could do 2, 3-way ANOVAs, but I would mainly like to see what happens to the model when fit to this data.

What is required for the model to capture the post-tetanus VEP changes? What if, for example, you held constant the ss->sp, ss->si, and sp->dp conducatnces? Would the model still be able to fit the data by allowing other parameters to vary? If so, does this mean that the choice of which variables to fix has a strong impact on the results?

Are there significant differences in any of the parameters when comparing the early and late conditions (in addition to comparing each to the pre-tetanus condition)?

Minor comments

Make sure layer/cell acronyms are defined

"Thus, providing the basis for the development of biologically plausible computational models of the canonical microcircuit that estimate the neural causes of EEG signal."

This sentence has no subject

"potentially as a result LTP occurring"

should be

"potentially as a result of LTP occurring"

"Where potentiation 30 minutes post-tetanus onward is likely to predominantly reflect E-LTP, and may contain some products of the conversion to L-LTP (Sumner et al., 2020; Sumner et al., In Press)."

Sentence fragment

"and a single pool on inhibitory"

should be

"and a single pool of inhibitory"

Reviewer #2: The authors present a validation of a thalamocortical model of the microcircuit structure of visual cortex, under the assumption that the observed changes in visual evoked potential are directly correlated with the columnar organization and functionality changes of specific excitatory and inhibitory neurons, in basal photic stimulation and post photic-tetanus conditions.

This study is interesting since photic induced VEP-LTP, can be used as landmark on neurological disorders and pharmacological treatment, as previously shown by the same authors (Summer, 2020). But the question remains about if those changes are also reflected at the neuronal network level and this paper provided some evidence in that direction.

However, there are some issues to be clarified before the publication.

The authors enthusiastically state that the evidence from previous work (Heynen and Bear, 2001(rat); Huang, 2014(monkey), which does not use photo stimulation) confirms the hypothesis proposed by them. At best this statement represents only a coincidence of cortical sinks and should not be taken as a validation of the authors' experimental results under any circumstances. A clearer discussion of this point is required.

A proper supporting paper are required to show the relevance of high-frequency visual stimulation to induce long lasting effect at cortical LTP, a unpublish paper should not be cited as supporting evidence across several section of the paper: “Sumner, Spriggs, Muthukumaraswamy, & Kirk, In Press”.

Minor points.

Please check the references section, some have the DOI annotated as URL and others not. Also there is missing information in some of them, for example: “Adams, N. E., Hughes, L. E., Phillips, H. N., Shaw, A. D., Murley, A. G., Nesbitt, D., . . . Rowe, J. B. (2020). GABAergic dynamics in human frontotemporal networks confirmed by pharmacomagnetoencephalography. Journal of Neuroscience, (XXXXX)?”

Reviewer #3: This manuscript is entitled: “Modelling thalamocortical circuitry shows visually induced LTP changes laminar connectivity in human visual cortex”. It builds on recent work for the development of biologically plausible generative models of EEG data, enabling to relate experimental modulations of scalp recordings to changes in laminar connectivity parameters at the microcircuit level. These Dynamic Causal Models (DCMs) have been extensively developed and refined over the last 10 years or so and represent an important paradigm shift to test mechanistic hypothesis, in health and disease, and evaluate treatments, non-invasively.

Precisely, this paper intends to contribute to this endeavour by bridging the gap between invasive studies in animal models and human in vivo essays. Therefore, it focuses on visually induced LTP changes using a previously published EEG tasks in adult volunteers. Part of the data presented here (female subjects) have been published previously.

Both for its objectives and for its original methodology, this paper is of importance to the field.

However, to fully validate the approach and the obtained findings, a crucial step is to demonstrate the identifiability of the proposed model as well as to convincingly show that the identified model is indeed plausible. In that respect, I have several questions and concerns. I have no doubt the authors can answer them which I believe will make the paper quite stronger and of interest to a broader readership.

Main comments

The authors have extended the microcircuit model previously published to cover their needs, given knowledge from animal studies. This makes perfect sense. However, those complex dynamic models have many parameters and especially when extended like here, they call for a careful validation. In particular:

- EEG is known to have a poor spatial resolution (namely w.r.t. MEG). It might well be that the well-motivated temporal constraints entailed by the model structure and interactions are sufficient to provide a plausible explanation for the observed experimental modulations of scalp EEG data. However, it might also be that a very large repertoire of alternative plausible and very different (possibly simpler) models would fit the data just as well. I am not suggesting here that the authors should consider an infinite model space. Some important factors/dimensions should be considered and investigated: for instance, is the thalamus compartment needed? (having in mind that activity in the thalamus is not directly ‘seen’ by EEG).

- Similarly, Bayesian Model Selection could further be used to formally test about the ability to estimate certain parameters, e.g. those who have been found to vary but whose modulations were not predicted by previous findings in animal models.

- In that respect, a few simulations that would show the difference in the repertoire of dynamics that the alternative models can cover would be highly informative (e.g. a model with an explicit vs. implicit thalamus compartment; or a model with vs without a non-linear effect). In line with recommendations in terms of model falsifiability [1].

- A related issue is the choice that the authors made for the data to be fitted by the DCMs. Instead of directly fitting the scalp data as commonly done with DCM for ERPs (that is fitting all sensors or their main spatial and temporal modes), the authors proceeded in two steps by first source-reconstructing the activity in a specific ROI of the visual cortex and fitted this reconstructed activity. Why ? I wonder if this may come with a critical loss of information ?

- Actually, the curves on Figure 4 tends to support this claim, as one can notice that the reconstructed dynamics in the chosen source do not exhibit all the experimental effects reported at the scalp level. In particular, there seems to be no effect at the N1 latency (which b.t.w seems to be shifted/postponed compared to the one at the scalp level).

- Surprisingly, very few information is provided regarding the source reconstruction step. What kind of anatomical model was considered (individual or template anatomy) ? How did you define the ROI location for each individual? What temporal window did you consider to apply MSP? This is a critical issue (in relation to the previous comment), since MSP assumes stationarity over the whole inverted data time window. This contrasts with the aim here, with DCM, to capture the finesse of dynamics in the visual system.

Also, what data did you reconstruct exactly (all conditions together at once, the difference between two conditions…?). And did you make sure that the considered ROI corresponded, for each subject, to a source that was explaining enough of the variance of the scalp response?

- Finally, you should report the individual Free Energies or Bayes Factors so that one can appreciate how your BMS results do reproduce over subjects and that the FFX approach is indeed justified.

Minor comments

- How many trials per subject were rejected following the described pre-processing procedure ?

- The authors used FCz as the reference. Usually an electrode that does not capture brain signals is preferred. Please comment and justify.

- In the methods section, ERP analysis paragraph, please specify explicitly the factors of the 3-way ANOVA.

- What do you mean by: “Simple effects analyses were carried out as appropriate” ?

- It would be helpful to show the spatio-temporal clusters that came out as significant in the ERP analysis.

- What software did you use for the RM-MANOVA ?

- A few typos:

• Page 11: “…, and a single pool OF inhibitory …”

• Better use Observation model everywhere instead of Observational model

• Page 19: “…the increase in variance causeD by…”

- I found some sentences a bit hard to read, including the title! Here are a few other examples:

“While less well documented the model will allow for the inhibitory changes to occur, potentially as a result LTP occurring on inhibitory interneurons.”

“In the first, pre-tetanus condition, both stimuli were presented in a random order 240 times at 1 Hz for 34.8 ms and takes approximately 8 minutes”

“The purpose of the ERP analysis was to confirm modulation of the ERP had occurred and so was

conducted solely to demonstrate evidence of VEP modulation post-tetanus”

“For this analysis we constrained the parameters that were allowed to vary (during model fitting)

according to the Douglas and Martin (2004) canonical microcircuit, and included all of the additional

parameters within layers II/III and IV, so this included … and meant the rest of the parameters were fixed”

Reference

[1] S. Palminteri, V. Wyart, et E. Koechlin, « The Importance of Falsification in Computational Cognitive Modeling », Trends Cogn. Sci., vol. 21, no 6, p. 425‑433, juin 2017, doi: 10.1016/j.tics.2017.03.011.

**Have all data underlying the figures and results presented in the manuscript been provided?**

Reviewer #1: **No: **I see a statement that the data are available upon request. They have not been provided here.

Reviewer #2: Yes

Reviewer #3: Yes

PLOS authors have the option to publish the peer review history of their article (what does this mean?). If published, this will include your full peer review and any attached files.

Reviewer #1: No

Reviewer #2: **Yes: **Carlos Rozas

Reviewer #3: No
---

## [Decision Letter · Decision Letter 1]

5 Oct 2020

Dear Dr Sumner,

We are pleased to inform you that your manuscript 'Modelling thalamocortical circuitry shows that visually induced LTP changes laminar connectivity in human visual cortex' has been provisionally accepted for publication in PLOS Computational Biology.

Best regards,

Hugues Berry

Associate Editor

PLOS Computational Biology

Daniele Marinazzo

Deputy Editor

PLOS Computational Biology

Reviewer's Responses to Questions

**Comments to the Authors:**

Reviewer #1: The authors have addressed all of my concerns and I'm happy to recommend this manuscript for publication.

Reviewer #2: The authors address properly my request of changes in their manuscript and now is in a better shape for publication

Reviewer #3: I have been very much satisfied by the authors thorough responses to my comments and their effort to usefully complement their manuscripts. I do believe the presentation of their really nice work has now become more convincing and accessible.

My minor comment on the stationnarity of the (MSP) source reconstruction step could have led to a short warning sentence added to the text but this is clearly not mandatory.

I have not written down but spotted a few typos here and there in the text that has been added, a last careful reading will enable the authors to correct for these.

Congratulations for this important contribution.

**Have all data underlying the figures and results presented in the manuscript been provided?**

Reviewer #1: Yes

Reviewer #2: Yes

Reviewer #3: Yes

PLOS authors have the option to publish the peer review history of their article (what does this mean?). If published, this will include your full peer review and any attached files.

Reviewer #1: No

Reviewer #2: **Yes: **Carlos Rozas

Reviewer #3: No

---

## [Editor Report · Acceptance letter]

18 Jan 2021

PCOMPBIOL-D-20-00468R1 

Modelling thalamocortical circuitry shows that visually induced LTP changes laminar connectivity in human visual cortex

Dear Dr Sumner,

I am pleased to inform you that your manuscript has been formally accepted for publication in PLOS Computational Biology. Your manuscript is now with our production department and you will be notified of the publication date in due course.

With kind regards,

Jutka Oroszlan
